# Advanced Image Forensics: Detecting Tampered and AI-Generated Images with Adversarial Learning

## Abstract

Detecting image tampering and Artificial Intelligence Generated Images are vital challenges in the fields of computer vision. The primary difficulty in identifying tampered images lies in uncovering minute evidence of manipulation, while AIGC image detection struggles with the increasingly lifelike quality of generated images. Existing solutions often focus on either tampered or AIGC images, yet both can coexist in various contexts. To address this limitation, we propose an advanced framework for social media image forensics that utilizes adversarial learning to identify both tampered and AIGC images. This framework employs a tri-branch architecture that combines generative adversarial learning with deep neural networks, effectively identifying both tampered and AI-generated content. We validated our framework using public image tampering datasets, a specialized AIGC image dataset, and a custom tampered AIGC image dataset . Experimental outcomes demonstrate that our framework significantly enhances accuracy by approximately 20%, 30%, and 20% for tampered, AI-generated, and tampered AIGC images, respectively. Code will be available on request.

## 1 Introduction

The rapid proliferation of social media platforms has led to a massive daily influx of information. Among these, images play a crucial role in information dissemination. However, with the advancement of editing tools and the development of sophisticated image generation technologies, tampered images have started to spread widely on social media, posing significant challenges for authenticity verification. As pointed out by Zhuang et al. (2023), image tampering detection is an essential task in computer vision, aiming at identifying and localizing modifications in images to ensure their authenticity. Visually realistic tampered images, often indistinguishable to the naked eye, can be maliciously circulated online, leading to societal misinformation, political manipulation, and social crises. Therefore, detecting tampered images is critical to mitigate the negative impact of false information. With the rapid development of AI-generated content (AIGC), powerful image editing models have provided a breeding ground for convenient image tampering, blurring the boundaries between true and forged images. Although it has facilitated the work of photographers and illustrators, Zhang et al. (2024) note that AIGC editing methods have also led to an increase in malicious tampering and illegal thefts. The authenticity of images in social media is difficult to guarantee, leading to problems such as rumor storms, economic losses, and legal concerns, as emphasized by Xu et al. (2024). Early approaches to image tampering detection heavily rely on manually designed features to identify inconsistencies and alterations in images. Techniques such as deploying Color Filter Array (CFA) correlations Popescu & Farid (2005), segmenting images into small blocks for similarity comparison Luo et al. (2006), analyzing frequency domains for unnatural variations Li et al. (2007), and utilizing Scale-Invariant Feature Transform (SIFT) to address geometric and illumination distortions Pan & Lyu (2010) are some of the initially employed methods. These methods aim to detect specific anomalies introduced during the tampering process. However, despite achieving some successes, these traditional approaches are constrained by the limitations of manually designed features. They require expert knowledge and are not robust enough to cope with the increasing complexity of tampering techniques used in modern image forensics.

The advent of deep learning has significantly advanced the field of image tampering detection, enabling automated feature extraction and improved accuracy. Convolutional Neural Networks (CNNs) extract abstract features through multi-level convolution and pooling operations, thus achieving better image classification and recognition. Examples include two-stage architectures based on R-CNN Yang et al. (2020), PSCC-Net Liu et al. (2022b), and MVSS-Net Dong et al. (2022). These deep learning-based methods have shown promising results in automating the detection process, enhancing the robustness of tampering detection, and addressing the shortcomings of traditional manual feature-based methods. The rapid advancement of Generative Adversarial Networks (GANs), generative algorithms, and pre-trained models has significantly enhanced the quality and diversity of AI-generated content (AIGC). The GAN architecture proposed by Goodfellow et al. (2014) is based on a core adversarial mechanism between a generator and a discriminator. The generator, using random noises as inputs, iteratively optimizes generated images to deceive the discriminator with their realism. The discriminator, meanwhile, attempts to distinguish between generated and real images with high precision. This adversarial training between generation and discrimination substantially improves the realism of images generated by the model. Different GAN-based AIGC generation algorithms have been developed, such as ProGAN Gao et al. (2019), StyleGAN Karras et al. (2019), and CycleGAN Zhu et al. (2017). These focus on image attributes, transformation mechanisms, or training processes for the realism of generated images. These advancements significantly improve the plasticity and realism of generated images while posing substantial technical challenges for traditional CNN-based tampering detection methods. Researchers have thus shifted to exploring GAN-based detection techniques Huang et al. (2022), aiming to improve detection capabilities in identifying generated images and addressing the complexities of modern image generation technologies. Xiao et al. (2023) show that GANs along with CNNs can enable pixel-level image detection, offering a more granular analysis of image tampering. This significantly improves detection sensitivity and allows for more detailed annotations of tampered regions, equipping the detection system with heightened sensitivity to subtle modifications. GANs can also work with dual-stage attention models for image tampering detection, as demonstrated by Islam et al. (2020). The attention mechanism is incorporated in both the first and second stages of the generator, allowing the model to integrate information effectively during detection and localization processes. This has achieved notable success in detecting copy-move forgeries, enabling the model to accurately identify and localize tampered areas, with particular advantages in detecting complex and detailed forgeries. By introducing attention mechanisms across different stages of the generator, the model is able to conduct more refined analyses of local image features, thereby achieving higher detection accuracy.

Despite advancements in tampering detection and AIGC detection, existing methods typically focus on either task. However, there exist both tampered and AI-generated images in many scenarios, making it infeasible to apply an individual image detection method focusing on a single task. The increasing realism of AI-generated images and the subtle traces of tampering demand a unified framework capable of detecting both types of image modifications. Therefore, in this work, we introduce a novel framework for social media image forensics based on adversarial learning, capable of detecting both tampered and AI-generated images, as illustrated in Figure 1. The proposed framework employs a tri-branch structure, integrating generative adversarial learning and deep neural networks to effectively detect

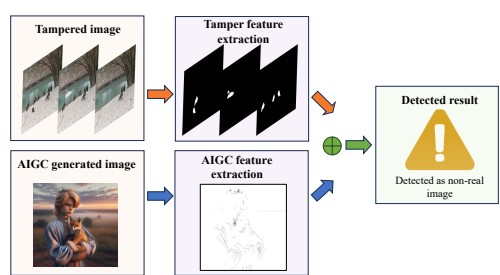

Figure 1: Overview of the proposed unified framework for detecting both tampered and AI-generated images. The system extracts features from tampering and AIGC branches, integrates them, and produces a final non-real image detection result.

both tampered and AIGC images. Our contributions are summarized as follows:

- We address the critical challenge of simultaneously detecting tampered and AI-generated images, where most methods are designed to handle only one of these tasks. By combining these two tasks into a unified framework, we pioneer a holistic approach to image foren-

sics, capable of addressing the increasingly complex scenarios presented by social media platforms.

- Our proposed framework employs a novel tri-branch discriminator architecture, which integrates edge-based, noise-based, and color-based feature extraction mechanisms. This design not only enables the accurate detection of general tampering and AI-generated content but also allows the model to effectively identify tampered AIGC images.

- We construct a dedicated dataset covering tampered images, AI-generated images, and tampered AI-generated images, enabling thorough validation across diverse scenarios. Through rigorous evaluation on multiple benchmark datasets and our custom dataset, our framework consistently outperforms state-of-the-art methods, achieving up to 20–30% higher accuracy in challenging scenarios. The model demonstrates exceptional robustness, even under noisy and distorted conditions.

## 2 RELATED WORK

### 2.1 IMAGE TAMPERING DETECTION METHODS

Traditional image tampering detection methods have relied on manually designed features to identify alterations in images. According to Popescu & Farid (2005), Color Filter Array (CFA) correlations can be used to expose digital forgeries by leveraging the regular pattern introduced by camera sensors. As described by Luo et al. (2006), segmenting images into small blocks for similarity comparison is effective in detecting copy-move forgeries. Li et al. (2007) analyzed the frequency domain to identify unnatural variations, while Pan & Lyu (2010) utilized Scale-Invariant Feature Transform (SIFT) features to detect geometric and illumination inconsistencies, which are common signs of manipulation. These traditional methods target specific anomalies introduced during the tampering process but require expert knowledge and are not robust enough to cope with increasingly complex modern techniques. With the development of deep learning, feature extraction has been automated, significantly improving the detection accuracy of tampered images. Yang et al. (2020) proposed a two-stage R-CNN architecture simulating a coarse-to-fine detection process, first identifying potential tampered regions and then refining the detection. Liu et al. (2022b) introduced a neural network that extracts cross-connected features from global to local scales, combining classification and convolutional networks to detect tampering artifacts at multiple scales and enhance robustness. Dong et al. (2022) developed a multi-perspective, multi-scale supervised tampering detection network balancing sensitivity and efficiency. Both Chen et al. (2022) and Guo et al. (2023) incorporated hierarchical feature extraction and attention mechanisms to improve detection accuracy. Recently, attention mechanisms have been widely applied in image tampering detection to capture subtle differences between regions. Zhao et al. (2021) integrated attention into CNNs to focus on tampered regions, enhancing accuracy by highlighting informative parts of the image. Adversarial learning has also been explored: Liu et al. (2023) surveyed adversarial learning-based frameworks for facial attribute manipulation detection, and Fang & Stamm (2023) developed EXIF-GAN to generate realistic tampered images, which are then used to attack image slicing and localization methods based on Siamese networks.

### 2.2 AI-GENERATED IMAGES DETECTION METHODS

The GAN architecture, as proposed by Goodfellow et al. (2014), is based on a core adversarial mechanism between a generator and a discriminator. Since 2017, GANs have revolutionized image generation through adversarial training between the generator and the discriminator. Brock et al. (2018) introduced BigGAN, incorporating orthogonal regularization within the generator and latent truncation to balance realism and diversity. Karras et al. (2017) proposed ProGAN with progressive growth and smooth transition, improving stability and quality. Diffusion models have emerged since 2021, excelling in producing high-quality facial and semantic images. Dhariwal & Nichol (2021) applied GAN optimization techniques to diffusion models, and Rombach et al. (2022) proposed latent-space diffusion for reduced computation and high-quality image generation. Nichol et al. (2021) introduced class-guided diffusion for controllable generation without retraining. In recent years, commercial APIs such as Stable Diffusion, Midjourney, and DALL-E 2 Rombach et al. (2022) have enabled users to create high-quality images from text without specialized knowledge,

expanding the accessibility of generative technology. In the field of detecting AI-generated images, early work trained binary classifiers to distinguish real from generated images. Wang et al. (2020) improved robustness by applying Gaussian blur and JPEG compression augmentations to training datasets. However, while effective on known generators, such methods performed poorly on unknown generators. To address this, Jeong et al. (2022) found that ignoring handcrafted frequency features could enhance generalization across GAN models, while Tan et al. (2023) transformed images into gradient maps as artificial features for classification. For diffusion models, Wang et al. (2023) proposed detection based on reconstructing inputs with pre-trained diffusion models and measuring the reconstruction error, and Ojha et al. (2023) extracted features from pre-trained models. Although various detection techniques exist for image tampering and AI-generated images, only few can concurrently detect both, especially in cases where tampered and AI-generated content coexist in the same image.

## 3 PROPOSED METHOD

In this work, we propose an innovative framework for detecting both image tampering and AI-generated images using a tri-branch discriminator structure paired with a generator, as shown in Fig. 2. The three branches respectively handle AI-generated image detection, edge-based tampering detection, and noise-based tampering detection. Features extracted from the input image are processed by the generator, and the outputs are analyzed by the discriminators to produce both tampering and AI-generated content detection results. The individual losses from each branch are combined into a single adversarial loss to optimize the generator during training.

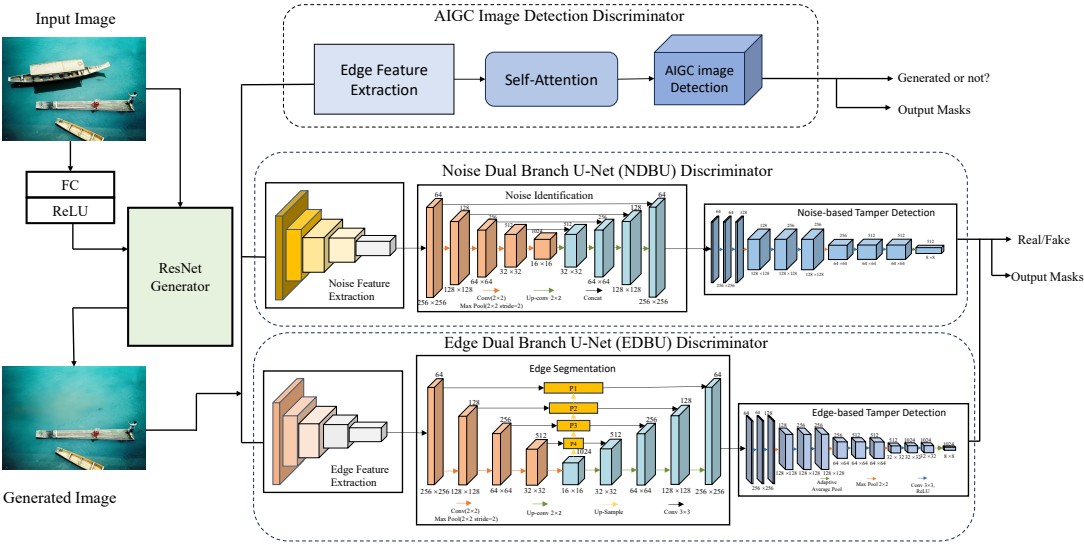

Figure 2: Proposed tri-branch discriminator and generator network

### 3.1 GENERATOR

The generator component of our framework is pivotal in synthesizing realistic images that emulate both tampered and AI-generated images. This synthesis is integral to the adversarial process, where the generator and discriminator engage in a continuous competition to enhance their performance. The generator's objective is to produce images indistinguishable from authentic ones, thereby challenging the discriminator to correctly differentiate between real and synthetic images.

The generator $G$ can be expressed as:

$$\hat{I}_t = G(I, z) \tag{1}$$

where $I$ represents the input image and $z$ denotes relevant edge, noise, and color features extracted from the input image. The generator learns to produce images $\hat{I}_t$ that the discriminator branches find indistinguishable from real images. The adversarial loss is defined as:

$$\mathcal{L}_{\text{adv}} = \mathbb{E}_{I \sim p_{\text{data}}(I)} \left[\log D(G(I, z))\right] + \mathbb{E}_{z \sim p_z(z)} \left[\log \left(1 - D(G(z))\right)\right] \tag{2}$$

where $\mathcal{L}_{\text{adv}}$ is the adversarial loss, $D$ represents the discriminator's probability output for an image being real, and $G(z)$ denotes the image generated from the input image and the relevant edge, color, and RGB features $z$.

Complementing this, the reconstruction loss ensures that the generated images preserve important structural and semantic information from the input images, which is critical for maintaining the realism. The reconstruction loss $\mathcal{L}_{\text{rec}}$ can be defined as:

$$\mathcal{L}_{\text{rec}} = \mathbb{E}_{I \sim p_{\text{data}}(I)} \left[\|I - G(I, z)\|_1\right] \tag{3}$$

The overall loss function for training the generator is a weighted sum of the adversarial loss and the reconstruction loss, as shown:

$$\mathcal{L}_G = \mathcal{L}_{\text{adv}} + \lambda_{\text{rec}} \mathcal{L}_{\text{rec}} \tag{4}$$

Here, $\mathcal{L}_G$ represents the total loss guiding the generator's training process. The term $\lambda_{\text{rec}}$ is a hyper-parameter that adjusts the balance between the adversarial loss $\mathcal{L}_{\text{adv}}$ and the reconstruction loss $\mathcal{L}_{\text{rec}}$. By tuning $\lambda_{\text{rec}}$, we control the emphasis placed on ensuring the generated images not only deceive the discriminator but also closely resemble the original images in terms of structure and semantics. This balance is crucial for achieving high-quality, realistic image generation.

### 3.2 TAMPERING DETECTION BRANCH

The tampering detection branch of our framework is designed to identify and localize image manipulations by leveraging two specialized discriminators: the EDBU discriminator and the NDBU discriminator. These components are tailored to capture and analyze distinct features that are indicative of tampering, ensuring a comprehensive detection approach.

The EDBU discriminator is centered around edge feature extraction and tampering detection. Initially, the generated image undergoes processing in the edge feature extraction module, which uses small convolutional kernels to capture fine-grained edge details. Following edge feature extraction, the data are passed through a U-Net structure enhanced with a feature Pyramid network (FPN) for multi-scale edge segmentation. The segmented edge features are then processed through an edge-based tamper detection layer.

The edge segmentation process is mathematically represented as:

$$E(x, y) = \text{Conv}_{\text{edge}}(\text{FPN}(U - \text{Net}(\text{EdgeFeatures}))) \tag{5}$$

The tampering likelihood score for edges is calculated as:

$$S_{\text{edge}}(x, y) = \sigma(w_e \cdot E(x, y) - \tau) \tag{6}$$

where $S(x, y)$ represents the sigmoid decision score at the pixel location $(x, y)$.

The NDBU discriminator is optimized for analyzing noise features. The process begins with an SRM filter applied to the generated image to extract frequency domain noise features. After extraction, the noise features are refined through a noise feature extraction module and processed via a classical U-Net structure for noise identification. The refined features are then passed through a noise-based tamper detection layer.

The noise detection process is mathematically represented as:

$$N(x, y) = \text{Conv}_{\text{noise}}(U_{\text{Net}}(\text{NoiseFeatures})) \tag{7}$$

The outputs from both the EDBU and NDBU discriminators are integrated to create a detailed tampering likelihood map:

$$S(x, y) = \sigma(w_e \cdot E(x, y) + w_c \cdot C(x, y) - \tau) \tag{8}$$

The custom loss function is defined as:

$$\mathcal{L} = -\frac{1}{N} \sum_{x,y} \Big[ P(x,y) \log(S(x,y)) +$$

$$(1 - P(x,y)) \log(1 - S(x,y)) \Big] + \lambda \Big( \|w_e\|_2^2 + \|w_c\|_2^2 \Big) \tag{9}$$

In this equation, the loss function $\mathcal{L}$ is a regularized binary cross-entropy loss that measures the difference between the predicted scores $S(x,y)$ and the ground truth labels $P(x,y)$, with $N$ being the total number of pixels. The term $\lambda$ controls the regularization strength.

## 3.3 AIGC IMAGES DETECTION BRANCH

The AIGC images detection branch is designed to distinguish between images generated by artificial intelligence and those captured by traditional cameras. It combines color feature extraction, attention mechanisms, and convolutional processing to identify unique characteristics of AI-generated images. The overall architecture is illustrated in Fig. 3.

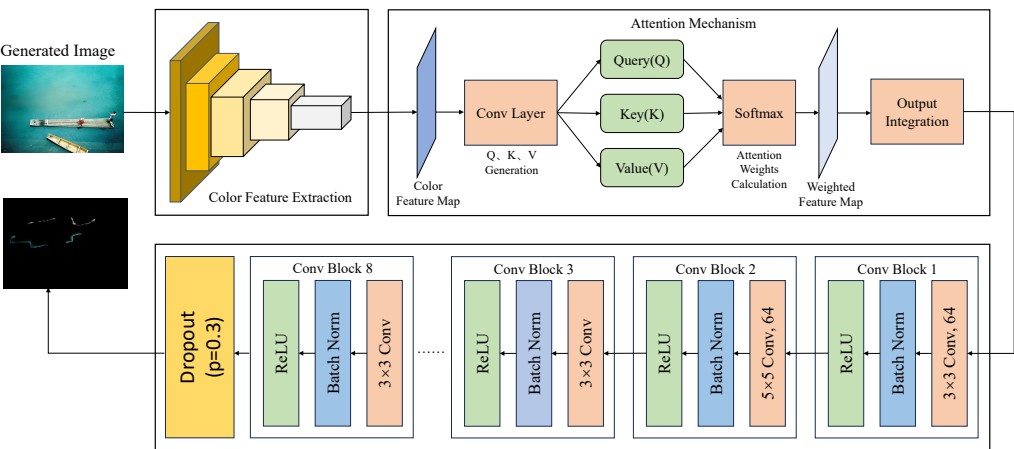

Figure 3: The architecture of the AIGC images detection branch, illustrating the process of feature extraction and the generation of a probability map to assess the likelihood of AI-generated images

First, color features are extracted from the input image to capture distinctive distributions and transitions that differ from real images. These features are then processed by an attention mechanism, which focuses on the most informative regions of the feature map to enhance subtle artifact detection. The attended features are refined through convolutional blocks that integrate spatial and spectral characteristics.

The branch outputs a probability map estimating the likelihood of each pixel being part of AI-generated content:

$$P_{\text{AIGC}}(x,y) = \sigma(F_{\text{AIGC}}(x,y)) \tag{10}$$

Training employs a binary cross-entropy loss to guide accurate detection:

$$\mathcal{L}_{\text{AIGC}} = -\frac{1}{M} \sum_{i=1}^{M} \Big[ y_i \log(p_i) + (1 - y_i) \log(1 - p_i) \Big] \tag{11}$$

The detailed formulas for color feature extraction, the attention mechanism, and intermediate feature transformations are provided in Appendix A.4.

## 3.4 FEATURE FUSION

The feature fusion process integrates the outputs from different discriminators to enhance the detection of both tampered and AI-generated images. Outputs from the EDBU and NDBU discriminators are combined for tampering detection, while the AIGC images detection discriminator provides AI-generated image results.

During training, the combined adversarial loss is calculated as:

$$L_{\text{adv}} = \alpha \cdot L_{\text{edge}} + \beta \cdot L_{\text{noise}} + \gamma \cdot L_{\text{AIGC images}} \tag{12}$$

where $\alpha$, $\beta$, and $\gamma$ are weights balancing the contributions of each loss component.

The final fusion function integrates the outputs from the three branches:

$$F(x,y) = \alpha \cdot T_E(x,y) + \beta \cdot T_N(x,y) + \gamma \cdot A(x,y) \tag{13}$$

These weights are optimized during training and regularized to maintain balance. Detailed parameter explanations and training strategies are provided in Appendix A.5.

## 4 EXPERIMENTS AND RESULTS

**Datasets** To evaluate the performance and robustness of our proposed framework, we conduct experiments on tampered datasets, GAN-generated datasets, and their combinations. The tampered datasets include CASIA V2, as described by Dong et al. (2013), and the IDM Dataset introduced by Ren et al. (2022). The GAN-generated datasets include images produced by ProGAN Gao et al. (2019), StyleGAN Karras et al. (2019), and CycleGAN Zhu et al. (2017). In addition, we construct a combined dataset containing both tampered and AI-generated images. The detailed configurations of these datasets are provided in Appendix A.7.

In addition, we introduce a new dataset specifically for evaluating tampering detection on AI-generated content, named **Tampered AIGC images Dataset**. This dataset is created by applying various tampering operations, including copy-move, splicing, inpainting, and adversarial tampering, to images generated by ProGAN, StyleGAN, and CycleGAN. The configuration of this dataset is shown in Table 4, which is used extensively in our experiments.

For evaluation, we adopt pixel-level metrics including accuracy, precision, recall, and F1-score for tampered region detection, and the same metrics at image level for AI-generated image detection.

To assess robustness, we further evaluate the framework under various noise and distortion conditions, including Gaussian noise, salt-and-pepper noise, JPEG compression, and blur. Full dataset details and additional tables are presented in Appendix A.7.

Figure 4: Configuration of the **Tampered AIGC Dataset**, obtained by applying tampering operations to images generated by ProGAN, StyleGAN, and CycleGAN, which is split into training and testing sets, with a combined version used in our experiments.

| Dataset | Train Set | Test Set | Total |
|---|---|---|---|
| ProG-Tampered | 8,000 | 2,000 | 10,000 |
| StyleG-Tampered | 8,000 | 2,000 | 10,000 |
| CycleG-Tampered | 8,000 | 2,000 | 10,000 |
| **Combined** | **24,000** | **6,000** | **30,000** |

**Impletion Details** We adopt batch normalization after each convolutional and deconvolutional layer to stabilize training and improve convergence. Dropout regularization with a rate of 0.5 is applied in the discriminator network to prevent overfitting. The Adam optimizer is used with a learning rate of 0.0002 and momentum parameters $(\beta_1, \beta_2) = (0.5, 0.999)$ to ensure stable and efficient convergence. The complete training and testing procedure, including the detailed algorithmic steps, loss computations, and fusion operations, is provided in Appendix A.6.

### 4.1 PERFORMANCE COMPARISON ACROSS DATASETS

We evaluate our method against several state-of-the-art tampering detection and AIGC detection methods on three types of datasets: tampered image datasets, GAN-generated datasets, and combined datasets.

Table 1: Performance comparison across tampered datasets, GAN-generated datasets, and combined datasets. ACC denotes accuracy and F1 denotes F1-score.

| Method | Tampered datasets | | | | GAN-generated datasets | | | | | | Combined datasets | | | | | |
| | CASIA V2 | | IDM | | ProGAN | | StyleGAN | | CycleGAN | | Tampered | | GAN | | Overall | |
| | ACC | F1 | ACC | F1 | ACC | F1 | ACC | F1 | ACC | F1 | ACC | F1 | ACC | F1 | ACC | F1 |
|---|---|---|---|---|---|---|---|---|---|---|---|---|---|---|---|---|
| MVSS | 0.78 | 0.77 | 0.79 | 0.76 | 0.41 | 0.40 | 0.42 | 0.41 | 0.32 | 0.34 | 0.77 | 0.78 | 0.28 | 0.26 | 0.31 | 0.33 |
| PSCC | 0.81 | 0.76 | 0.83 | 0.78 | 0.43 | 0.42 | 0.33 | 0.31 | 0.45 | 0.44 | 0.81 | 0.84 | 0.33 | 0.36 | 0.39 | 0.44 |
| CNNSpot | 0.46 | 0.51 | 0.53 | 0.49 | 0.63 | 0.61 | 0.64 | 0.62 | 0.55 | 0.53 | 0.38 | 0.32 | 0.66 | 0.61 | 0.57 | 0.59 |
| FreDect | 0.61 | 0.64 | 0.55 | 0.58 | 0.75 | 0.73 | 0.76 | 0.74 | 0.67 | 0.69 | 0.64 | 0.61 | 0.76 | 0.76 | 0.72 | 0.74 |
| GramNet | 0.65 | 0.61 | 0.61 | 0.64 | 0.77 | 0.75 | 0.78 | 0.76 | 0.79 | 0.77 | 0.63 | 0.61 | 0.80 | 0.74 | 0.74 | 0.76 |
| LNP | 0.70 | 0.70 | 0.75 | 0.74 | 0.81 | 0.79 | 0.81 | 0.80 | 0.83 | 0.81 | 0.73 | 0.71 | 0.81 | 0.79 | 0.80 | 0.78 |
| TruFor | 0.78 | 0.77 | 0.80 | 0.79 | 0.83 | 0.81 | 0.88 | 0.86 | 0.86 | 0.85 | 0.78 | 0.77 | 0.83 | 0.82 | 0.84 | 0.80 |
| DFVT | 0.72 | 0.70 | 0.74 | 0.72 | **0.85** | 0.81 | 0.91 | **0.91** | 0.88 | 0.86 | 0.74 | 0.72 | 0.84 | 0.82 | 0.83 | 0.81 |
| **Ours** | **0.85** | **0.84** | **0.86** | **0.88** | 0.84 | 0.82 | **0.92** | 0.90 | **0.89** | **0.87** | **0.84** | **0.82** | **0.86** | **0.84** | **0.85** | **0.83** |

On tampered datasets (CASIA V2, IDM), our approach achieves the highest accuracy of **0.85** and **0.86**, and F1-scores of **0.84** and **0.88**, respectively, outperforming the second-best method PSCC by up to 5% in F1-score. On GAN-generated datasets, our model attains **0.84/0.82** (ACC/F1) on Pro-GAN, **0.92/0.90** on StyleGAN, and **0.89/0.87** on CycleGAN, exceeding the best baseline (TurFor) by 0.03 in accuracy. On combined datasets, our framework achieves **0.84/0.82** on Tampered Combined, **0.86/0.84** on GAN Combined, and **0.85/0.83** on Overall Combined, consistently ranking first across all metrics. Additionally, subjective comparisons of several methods are shown in Figure 5.

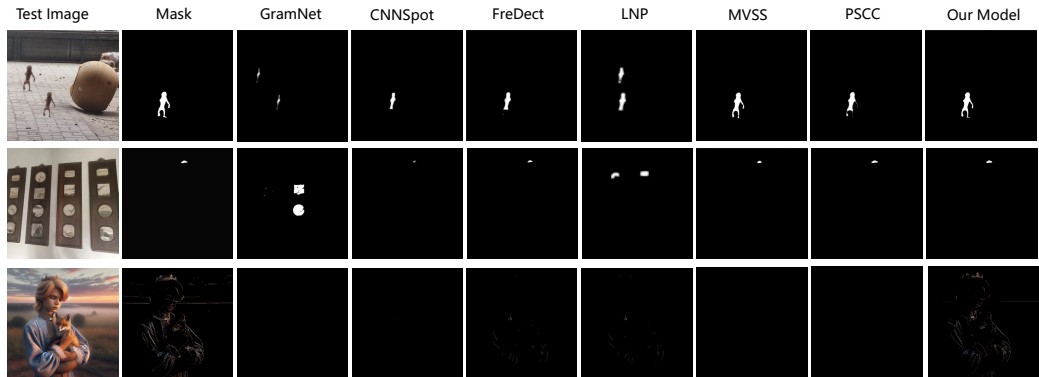

Figure 5: Visual comparison of different models on example datasets.

## 4.2 PERFORMANCE ON TAMPERED IMAGE DATASET

To evaluate the performance of our method on tampered AIGC images, we conduct experiments on the newly constructed **Tampered AIGC Dataset** and compare our results with several state-of-the-art tampering detection methods (e.g., MVSS Dong et al. (2022), PSCC Liu et al. (2022b)) and AIGC detection methods (e.g., CNNSpot Wang et al. (2020), FreDect Frank et al. (2020), GramNet Liu et al. (2020), LNP Liu et al. (2022a)). The results are summarized in Table 2.

Table 2: Performance on Tampered AIGC image Dataset

| Method | ProG-Tampered | | StyleG-Tampered | | CycleG-Tampered | | Combined | |
| | ACC | F1 | ACC | F1 | ACC | F1 | ACC | F1 |
|---|---|---|---|---|---|---|---|---|
| MVSS | 0.35 | 0.33 | 0.38 | 0.36 | 0.32 | 0.30 | 0.35 | 0.33 |
| PSCC | 0.40 | 0.37 | 0.43 | 0.40 | 0.38 | 0.35 | 0.40 | 0.37 |
| CNNSpot | 0.54 | 0.51 | 0.57 | 0.54 | 0.50 | 0.47 | 0.54 | 0.51 |
| FreDect | 0.62 | 0.59 | 0.65 | 0.62 | 0.58 | 0.55 | 0.62 | 0.59 |
| GramNet | 0.67 | 0.64 | 0.69 | 0.66 | 0.64 | 0.61 | 0.67 | 0.64 |
| LNP | 0.72 | 0.70 | 0.74 | 0.72 | 0.70 | 0.67 | 0.72 | 0.70 |
| **Ours** | **0.78** | **0.75** | **0.81** | **0.78** | **0.77** | **0.74** | **0.79** | **0.76** |

Our method consistently outperforms all compared state-of-the-art tampering and AIGC detection methods across all datasets. Specifically, on the ProG-Tampered dataset, our approach achieves an accuracy of 78% and an F1-score of 0.75, demonstrating strong detection performance. On the StyleG-Tampered dataset, the accuracy rises to 81% with an F1-score of 0.78, indicating excellent robustness. Similarly, on the CycleG-Tampered dataset, our method attains an accuracy of 77% and an F1-score of 0.74. When evaluated on the combined dataset, the model reaches an accuracy of 79% and an F1-score of 0.76. These results confirm the robustness and high effectiveness of our framework in detecting images generated by various GAN models, significantly surpassing existing methods in both accuracy and F1 measures.

### 4.3 ABLATION STUDIES

To analyze the contribution of each component in our framework, we conduct ablation studies on both the Overall Combined Dataset and the AIGC-Tampered Dataset. The results, presented in Table 3, show the impact of removing individual components on the performance.

Table 3: Ablation Study on the Effects of Different Components

| Method | Overall Combined Dataset | | AIGC-Tampered Dataset | |
|---|---|---|---|---|
| | ACC | F1 | ACC | F1 |
| Without AIGC Branch | 0.51 | 0.47 | 0.60 | 0.58 |
| Without Tamper Branch | 0.72 | 0.71 | 0.75 | 0.73 |
| Without Attention | 0.68 | 0.70 | 0.72 | 0.70 |
| Without Fusion | 0.70 | 0.66 | 0.71 | 0.69 |
| **Ours** | **0.85** | **0.83** | **0.89** | **0.88** |

Removing the AIGC detection branch leads to a significant drop in performance on both datasets, with the accuracy decreasing to 0.51 and 0.60, and the F1-score to 0.47 and 0.58, respectively. Similarly, removing the tampering detection branch reduces the accuracy to 0.72 and 0.75, and the F1-score to 0.71 and 0.73, respectively. When all components are included, our proposed framework achieves the best results, compared with model with partial components, with an accuracy of 0.85 and 0.89, and an F1-score of 0.83 and 0.88 on the Overall Combined Dataset and the AIGC-Tampered Dataset, respectively.

### 4.4 ROBUSTNESS EVALUATION ON NOISY AND DISTORTED DATA

To further validate the robustness of our framework, we conduct additional experiments by introducing various types of noises and distortions to the combined dataset. The results, shown in Table 4, indicate that our method achieves superior performance even under noisy and distorted conditions.

Table 4: Performance Comparison on Noisy and Distorted Datasets

| Method | Gaussian | Salt-Pepper | JPEG | Blur |
|---|---|---|---|---|
| MVSS | 0.22 | 0.27 | 0.22 | 0.26 |
| PSCC | 0.28 | 0.31 | 0.29 | 0.36 |
| CNNSpot | 0.49 | 0.52 | 0.38 | 0.51 |
| FreDect | 0.57 | 0.63 | 0.46 | 0.62 |
| GramNet | 0.66 | 0.57 | 0.55 | 0.68 |
| LNP | 0.68 | 0.70 | 0.65 | 0.66 |
| TBD-GAN (Ours) | **0.75** | **0.81** | **0.68** | **0.71** |

The results, shown in Table 4, indicate that our method achieves superior performance even under noisy and distorted conditions. Our approach consistently outperforms other state-of-the-art methods across all distortion types, with particularly large margins under Salt-and-Pepper noise and Gaussian noise. This demonstrates that the proposed tri-branch architecture, combined with multi-feature fusion, is able to capture robust discriminative cues that remain effective even when the visual quality of the input is heavily degraded.

## 5 CONCLUSION

In this paper, we propose a cutting-edge framework for image forensics based on adversarial learning, designed to simultaneously address the challenges of detecting image tampering and AI-

generated images. This innovative framework adopts a tri-branch structure, cleverly integrating generative adversarial learning and multi-feature extraction capabilities of deep neural networks, while achieving an effective fusion of image tamper detection and AIGC image detection. Through rigorous evaluation on multiple benchmark datasets and our custom dataset, our framework consistently outperforms state-of-the-art methods, achieving up to 20–30% higher accuracy in challenging scenarios. Our model effectively addresses the challenges posed by the increasing realism of AI-generated images and the subtle traces of tampering, providing a solution for social media image forensics. Additionally, we evaluate the robustness of our framework under noisy and distorted conditions. The results show that our method maintains high performance even when subjected to various types of noise and distortions.

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

# A APPENDIX

## A.1 FRAMEWORK OVERVIEW DETAILS

Figure 2 illustrates three distinct modules: the AIGC images detection discriminator, the Edge Dual Branch U-Net (EDBU) discriminator, and the Noise Dual Branch U-Net (NDBU) discriminator. The process begins with an input image undergoing preprocessing via Fourier convolution and the activation function to extract relevant features. These features, alongside the original image, are fed into a ResNet Generator, which produces a generated image. This image is then processed by the three discriminators. The AIGC images detection discriminator is tasked with identifying AIGC imagess by extracting color features and employing a series of convolutional blocks along with an attention mechanism. This module outputs the regions likely to be AI-generated, distinguishing them from real images.

Simultaneously, the EDBU discriminator focuses on detecting tampering by analyzing edge and RGB features. It involves edge segmentation followed by edge-based tampering detection, outputting regions suspected of being tampered with based on edge inconsistencies. On the other hand, the NDBU discriminator identifies tampering by examining noise features. It performs noise identification and applies noise-based tampering detection, highlighting regions likely tampered due to abnormal noise patterns.

The outputs from the EDBU and NDBU discriminators are combined to provide a comprehensive tampering detection result, while the AIGC images detection discriminator outputs the AI-generated content detection result. During training, the results from each branch are compared with the ground truth to compute individual losses, which are integrated to form a single adversarial loss. This loss is then backpropagated to the generator to refine its outputs. In the following sections, we detail each module's architecture and specific role within the overall framework.

## A.2 GENERATOR ADDITIONAL DETAILS

Our generator is adept at handling the complexities of both image tampering and AI-generated images, which makes it versatile and robust. It employs a GAN-based process consisting of a series of convolutional and deconvolutional layers that transform the input image along with extracted relevant edges, noises, and color features into a coherent and realistic output image. This iterative refinement process allows the generator to rectify any initial inconsistencies and enhance the overall quality of the generated images.

The adversarial loss, $\mathcal{L}_{\text{adv}}$, is calculated by evaluating two expectations: one over real images, $\mathbb{E}_{I \sim p_{\text{data}}(I)} [\log D(I)]$, which ensures the discriminator correctly identifies real images, and the other over generated images, $\mathbb{E}_{z \sim p_z(z)} [\log (1 - D(G(z)))]$, encouraging the generator to improve its outputs by maximizing the discriminator's error in distinguishing them from real images. Here, $D$ represents the discriminator's probability output for an image being real, while $G(z)$ denotes the image generated from the latent noise $z$.

Complementing this, the reconstruction loss ensures that the generated images preserve important structural and semantic information from the input images, which is critical for maintaining the realism.

By tuning $\lambda_{\text{rec}}$, we control the emphasis placed on ensuring the generated images not only deceive the discriminator but also closely resemble the original images in terms of structure and semantics. This balance is crucial for achieving high-quality, realistic image generation.

## A.3 ADDITIONAL DETAILS FOR TAMPERING DETECTION BRANCH

The edge features extracted by the EDBU branch are crucial for detecting subtle discontinuities or unnatural patterns introduced by manipulations. They are particularly effective in identifying manipulations like copy-move forgeries, where the misalignment of edges is a common occurrence. The integration of multi-scale edge information through the FPN enables comprehensive edge segmentation, allowing the detection of even the most subtle manipulations across various scales.

In the NDBU branch, the SRM filter extracts frequency domain noise features that often highlight variations indicative of tampering. This stage is essential for detecting anomalies in noise patterns, such as those caused by splicing or inpainting, where noise consistency is disrupted.

For the loss function in the tampering detection branch, the regularization strength $\lambda$ plays a crucial role in encouraging generalization by penalizing the complexity of the weights through their $L_2$-norms.

Finally, in our dual-discriminator design, smaller convolutional kernels in the edge branch enable detailed analysis of image boundaries and potential tampering artifacts, while larger kernels in the noise branch provide a broader understanding of noise distribution. This distinction enhances the model's ability to identify a wide range of tampering artifacts, ensuring robust and reliable detection.

## A.4 DETAILED DESCRIPTION OF THE AIGC IMAGES DETECTION BRANCH

The initial stage of the AIGC images detection branch focuses on extracting color features from the input image. AI-generated images often exhibit specific color distributions and transitions that differ from those in real images. This involves a series of convolutional layers that process the image to capture these transitions and patterns. These features are critical for identifying the unique color characteristics of AI-generated content. The extraction process can be expressed as:

$$F_{\text{color}} = \text{Conv}_{\text{color}}(I) \tag{14}$$

where $F_{\text{color}}$ represents the extracted color features and $\text{Conv}_{\text{color}}$ denotes the convolutional operations applied to the input image $I$.

The attention mechanism is a central component in the AIGC images detection branch. It starts by transforming the color feature map $F$ into the query $Q$, key $K$, and value $V$ matrices:

$$Q = W_q \cdot F, \quad K = W_k \cdot F, \quad V = W_v \cdot F \tag{15}$$

Similarity scores are computed:

$$S = Q \cdot K^T \tag{16}$$

These scores are scaled by $\sqrt{d_k}$ and normalized using a softmax function:

$$W_{(i,j)} = \frac{\frac{S_{(i,j)}}{\sqrt{d_k}}}{\sum_{i=1}^{H} \sum_{j=1}^{W} \exp \frac{S_{i,j}}{\sqrt{d_k}}} \tag{17}$$

The weighted feature map is obtained as:

$$F' = W \cdot V \tag{18}$$

This integrates the attention weights with the original feature values, emphasizing the most relevant regions while suppressing less important information. This focus helps capture intricate details and subtle variations, such as unnatural color transitions or edge inconsistencies.

The attended feature map $F'$ is then processed through additional convolutional layers to integrate spatial and spectral characteristics. It passes through a series of convolutional blocks, each consisting of a convolution layer, batch normalization, and a ReLU activation function, refining features and enhancing artifact detection.

By integrating color feature extraction, the attention mechanism, and convolutional processing, the AIGC images detection branch provides a robust solution for distinguishing between AI-generated and real images, leveraging both spatial and spectral domains for high precision.

### A.5 DETAILED DESCRIPTION OF THE FEATURE FUSION PROCESS

The feature fusion process in our framework is designed to integrate the outputs from different discriminators, enhancing the model's ability to detect both tampered and AI-generated images. The workflow involves combining the results from the EDBU discriminator and the NDBU discriminator to produce a comprehensive tampering detection output. Meanwhile, the AIGC images detection discriminator independently provides the AI-generated image detection result.

During training, each discriminator produces a specific loss: $L_{\text{edge}}$ from the EDBU discriminator, $L_{\text{noise}}$ from the NDBU discriminator, and $L_{\text{AIGC images}}$ from the AIGC images detection discriminator. These losses are crucial for guiding the learning process of the generator. The combined adversarial loss $L_{\text{adv}}$ is calculated by integrating these individual losses, ensuring that the generator is optimized to produce realistic images that can fool all three discriminators:

$$L_{\text{adv}} = \alpha \cdot L_{\text{edge}} + \beta \cdot L_{\text{noise}} + \gamma \cdot L_{\text{AIGC images}} \tag{19}$$

In this equation, $\alpha$, $\beta$, and $\gamma$ are weights that balance the contributions of each loss component. These weights are tuned to ensure that each aspect of the detection, including edge inconsistencies, noise patterns, and AI-generated artifacts, is appropriately addressed.

The total adversarial loss $L_{\text{adv}}$ is then back-propagated to the ResNet generator, refining its ability to generate images that are difficult for the discriminators to classify as tampered or AI-generated. This integrated approach leverages the complementary strengths of each discriminator, improving the overall robustness and accuracy of the model.

The fusion function $F(x, y)$ in our framework combines the outputs from the three branches: Edge Dual Branch U-Net (EDBU), Noise Dual Branch U-Net (NDBU), and the AIGC images detection branch, using a weighted sum:

$$F(x, y) = \alpha \cdot T_E(x, y) + \beta \cdot T_N(x, y) + \gamma \cdot A(x, y) \tag{20}$$

Here, $\alpha$, $\beta$, and $\gamma$ are learnable weights that control the contributions of each branch. These weights are optimized during training via backpropagation, jointly with the generator and discriminator parameters, and regularized with an $L_2$-norm term to maintain balance and prevent overfitting. The fusion process dynamically adapts to the varying importance of each branch.

## A.6 DETAILED IMPLEMENTATION AND TRAINING PROCEDURE

We use batch normalization layers after each convolutional and deconvolutional layer to stabilize the training process and improve the convergence speed. Batch normalization helps maintain the mean and variance of the activations within a reasonable range, which is crucial for training deep networks.

To prevent overfitting, dropout regularization is applied in the discriminator network. Dropout randomly sets a fraction of the activations to zero during training, forcing the network to learn redundant representations and improving generalization. The dropout rate is set to 0.5, meaning that 50% of the activations are dropped during training.

The optimization of the framework is performed using the Adam optimizer, which combines the advantages of both the AdaGrad and RMSProp algorithms, providing adaptive learning rates for each parameter. Specifically, the learning rate is set to 0.0002, and the momentum parameters $(\beta_1, \beta_2)$ are set to 0.5 and 0.999, respectively. These settings help achieve stable and fast convergence during training.

---

**Algorithm 1** TBD-GAN Training and Testing Procedure

---

**Input:** Input image $I$
**Output:** Estimated result
1: **for** each training step from 1 to $N$ **do**
2:     Extract Fourier features from $I$.
3:     Generate image $\hat{I}$ using the ResNet Generator $G$ with input features and noise $z$.
4:     Compute the noise-based tampering detection loss $L_{\text{noise}}$ using the NDBU Discriminator on $\hat{I}$.
5:     Compute the edge-based tampering detection loss $L_{\text{edge}}$ using the EDBU Discriminator on $\hat{I}$.
6:     Compute the AI-generated image detection loss $L_{\text{AIGC images}}$ using the AIGC images Detection Discriminator on $\hat{I}$.
7:     Compute the total adversarial loss $L_{\text{adv}} = \alpha \cdot L_{\text{noise}} + \beta \cdot L_{\text{edge}} + \gamma \cdot L_{\text{AIGC images}}$.
8:     Update the Generator $G$ using $L_{\text{adv}}$.
9: **end for**
10: **for** each testing image $I_{\text{test}}$ **do**
11:     Evaluate $I_{\text{test}}$ using the trained model $G$, $NDBU$, $EDBU$, and $AIGCimages$ Discriminators.
12:     Output the tampering likelihood map $T(x, y)$ and AI-generated probability map $A(x, y)$.
13:     Derive the final authenticity verdict using the fusion function $F(x, y)$.
14: **end for**

---

The training procedure for the framework is outlined in Algorithm 1. During each training step, Fourier features are extracted from the input image $I$. The ResNet generator $G$ then uses these features along with latent noise variables $z$ to produce generated images $\hat{I}$. Three distinct losses are computed: the noise-based tampering detection loss $L_{\text{noise}}$ using the NDBU discriminator, the edge-based tampering detection loss $L_{\text{edge}}$ using the EDBU discriminator, and the AI-generated content detection loss $L_{\text{AIGC images}}$ using the AIGC images detection discriminator. These losses are combined to form the total adversarial loss $L_{\text{adv}}$, which guides the training process.

In the testing phase, each image $I_{\text{test}}$ is assessed for authenticity using the trained model components $G$, $NDBU$, $EDBU$, and $AIGCimages$ discriminators. The outputs are a tampering likelihood map $T(x, y)$ and an AI-generated probability map $A(x, y)$. These outputs are combined using a fusion function $F(x, y)$ to derive the final authenticity verdict, ensuring robust detection of both tampered and AI-generated content.

## A.7 DATASET DETAILS AND ADDITIONAL TABLES

### A.7.1 DATASET CONFIGURATIONS

We utilize two main types of datasets: tampered datasets and GAN-generated datasets. The tampered datasets include CASIA V2 Dong et al. (2013), which contains 4,795 images (1,701 authentic and 3,274 tampered), and the IDM Dataset Ren et al. (2022), which has 5,000 tampered images generated using various sophisticated tampering techniques.

For the GAN-generated datasets, we use images produced by three state-of-the-art GAN architectures: ProGAN Gao et al. (2019), StyleGAN Karras et al. (2019), and CycleGAN Zhu et al. (2017). We generate 10,000 images with each of these GANs, resulting in a total of 30,000 GAN-generated images. We name these datasets ProG-10K, StyleG-10K, and CycleG-10K, respectively.

We also create a combined dataset including all images from both the tampered datasets and GAN-generated datasets, totaling 39,795 images.

Table 5: Detailed Description of Datasets Used in the Experiments

| Dataset | Category | Train Set | Test Set | Total |
|---|---|---|---|---|
| CASIA V2 | Tampered | 1,700 | 1,095 | 2,795 |
| IDM | Tampered | 3,000 | 2,000 | 5,000 |
| ProG-10K | GAN-gen | 8,000 | 2,000 | 10,000 |
| StyleG-10K | GAN-gen | 8,000 | 2,000 | 10,000 |
| CycleG-10K | GAN-gen | 8,000 | 2,000 | 10,000 |
| Combined | Mixed | 28,700 | 10,095 | 39,795 |

The pixel-level F1-score is defined as:

$$\text{F1-Score}_{\text{pixel}} = 2 \cdot \frac{\text{Precision}_{\text{pixel}} \cdot \text{Recall}_{\text{pixel}}}{\text{Precision}_{\text{pixel}} + \text{Recall}_{\text{pixel}}} \qquad (21)$$

where $\text{Precision}_{\text{pixel}}$ and $\text{Recall}_{\text{pixel}}$ are computed over all pixels in the tampered region.

### A.7.2 AIGC-TAMPERED DATASET CONSTRUCTION

The AIGC-Tampered Dataset is constructed using images generated by ProGAN, StyleGAN, and CycleGAN. Tampering operations include:

- **Copy-Move:** Copying a region within the image and pasting it elsewhere.
- **Splicing:** Replacing regions in an image with content from another image.
- **Inpainting:** Filling missing or removed parts of an image using image repair techniques.
- **Adversarial Tampering:** Introducing local modifications using adversarial attack methods.

We generate 10,000 images with each GAN model, resulting in 30,000 base images. 24,000 images (8,000 per GAN model) are used for training and 6,000 images (2,000 per GAN model) for testing.

### A.7.3 EVALUATION METRICS

Accuracy measures the proportion of correctly classified pixels across the entire image. For tampered regions, precision evaluates the proportion of predicted tampered pixels that are correct, while recall reflects the proportion of actual tampered pixels that are successfully identified. The pixel-level F1-score combines precision and recall as:

$$\text{F1-Score}_{\text{pixel}} = 2 \cdot \frac{\text{Precision}_{\text{pixel}} \cdot \text{Recall}_{\text{pixel}}}{\text{Precision}_{\text{pixel}} + \text{Recall}_{\text{pixel}}} \qquad (22)$$

The same metrics are applied at the image level for AI-generated content detection.

We also evaluate under different noise and distortion conditions: Gaussian noise, salt-and-pepper noise, JPEG compression, and blur.

A.8 DETAILED PERFORMANCE ANALYSIS

A.8.1 PERFORMANCE ON TAMPERED IMAGE DATASETS

We compare our method against several state-of-the-art tampering detection methods (e.g., MVSS Dong et al. (2022), PSCC Liu et al. (2022b)) and AIGC detection methods (e.g., CNNSpot Wang et al. (2020), FreDect Frank et al. (2020), GramNet Liu et al. (2020), LNP Liu et al. (2022a)) and Turfor(Guillaro et al. (2023)). The results, summarized in Table 1, indicate that our approach significantly outperforms existing methods in terms of F1-score and accuracy. Our results are highlighted in bold and the second-best results are underlined. From the table, it is evident that our method significantly outperforms other state-of-the-art methods across both CASIA V2 and IDM datasets. Our approach achieves the highest accuracy and F1-scores, demonstrating the superior robustness in detecting tampered images. While methods like MVSS and PSCC show the competitive performance, particularly on the IDM dataset, they are outperformed by our model. This highlights the effectiveness of our dual-branch U-net architecture in capturing both edge and color inconsistencies, as well as the advantages of integrating adversarial learning and multi-feature extraction. GramNet, which analyzes global texture features, achieves moderate results, but it does not match the accuracy and F1-scores of our approach. Similarly, traditional methods like CNNSpot and FreDect fall short in the performance, underscoring the impact of the attention mechanisms in our model that enhance detection by focusing on the most informative image regions. TruFor also achieves notable accuracy and F1-scores, ranking second overall in this category and confirming its strong capability in tampering detection. Overall, the comprehensive design of our framework provides a significant advantage in identifying subtle tampering artifacts across varied datasets.

A.8.2 PERFORMANCE ON GAN-GENERATED DATASETS

We compare our method against several existing tampering detection methods and AIGC images detection methods. From the table, we notice that traditional tampering detection methods perform poorly on generated image datasets, while our method achieves the highest performance. The training set consists of 8,000 images from ProGAN, 8,000 images from StyleGAN, and 8,000 images from CycleGAN. The test set consists of 2,000 images from ProGAN, 2,000 images from Style-GAN, and 2,000 images from CycleGAN. Table 1 shows that our method performs exceptionally well on GAN-generated datasets, achieving the highest accuracy (0.84, 0.92, and 0.89) and F1-scores (0.82, 0.90, and 0.87) across ProGAN, StyleGAN, and CycleGAN datasets. Traditional tampering detection methods like MVSS and PSCC exhibit significantly lower performance, highlighting their limitations in handling AI-generated images. GramNet and LNP, which are specifically designed for AIGC images detection, show competitive results but still fall short of ours. TruFor performs well in this setting, particularly on StyleGAN and CycleGAN, surpassing LNP and narrowing the gap with our method, which suggests its adaptability to synthetic content. The superior performance of our method can be attributed to its tailored architecture that effectively captures the unique characteristics of GAN-generated content. The attention mechanism in our model enhances the detection performance by focusing on the most informative regions, further improving the accuracy of AIGC images detection.

A.8.3 PERFORMANCE ON COMBINED DATASETS

We evaluate the performance of our method on the combined dataset against existing tampering detection and AIGC detection methods. The results, shown in Table 1, indicate that our method achieves the superior performance on all types of datasets. Our method consistently achieves the highest accuracy (0.84, 0.86, and 0.85) and F1-scores (0.82, 0.84, and 0.83) on tampered combined, GAN combined, and overall combined datasets, respectively. This demonstrates the effectiveness of our unified approach in handling both tampered and AI-generated images. Such traditional methods as CNNSpot and FreDect show the moderate performance, but are outperformed by our method due to their limited ability to generalize across different types of image manipulations. Methods specifically designed for either tampering detection or AIGC images detection, such as GramNet and LNP, perform well on their respective tasks but do not achieve the same level of performance on combined datasets. TruFor maintains strong results across all combined datasets, consistently ranking second overall, which reflects its balanced generalization ability across tampered and synthetic images. Our model's ability to integrate the multi-feature extraction and adversarial learning

allows it to effectively detect and distinguish between tampered and AI-generated images, providing a robust solution for image forensics.

## A.9 ABLATION STUDY

To analyze the contribution of each component in our framework, we conduct ablation studies on both the Overall Combined Dataset and the AIGC-Tampered Dataset. The results, presented in Table 3, show the impact of removing individual components on the performance. Removing the AIGC detection branch leads to a significant drop in performance on both datasets, with the accuracy decreasing to 0.51 and 0.60, and the F1-score to 0.47 and 0.58, respectively. This indicates the critical role of the AIGC branch in identifying AI-generated content, particularly for datasets containing subtle generative artifacts. Similarly, removing the tampering detection branch reduces the accuracy to 0.72 and 0.75, and the F1-score to 0.71 and 0.73, respectively. This highlights the importance of the tampering branch in identifying image manipulations, such as splicing and copy-move operations.

The attention mechanism also plays a vital role in our model, as its removal results in noticeable performance degradation (accuracy of 0.68 and 0.72, F1-score of 0.70 in both cases). By focusing on the most informative regions of the image, the attention mechanism enhances the model's ability to detect subtle tampering and AI-generated artifacts. Finally, removing the fusion component leads to a decline in accuracy to 0.70 and 0.71, and the F1-score to 0.66 and 0.69, respectively. The fusion component integrates features from both the AIGC branch and the tampering detection branch, enabling the model to leverage complementary information for more robust performance. When all components are included, our proposed framework achieves the best results, with an accuracy of 0.85 and 0.89, and an F1-score of 0.83 and 0.88 on the Overall Combined Dataset and the AIGC-Tampered Dataset, respectively. These results underscore the importance of each component and demonstrate how their integration enables the model to achieve state-of-the-art performance across diverse datasets.

## A.10 ROBUSTNESS STUDY

To further validate the robustness of our framework, we conduct additional experiments by introducing various types of noises and distortions to the combined dataset. The goal is to assess the performance of our model under challenging conditions and compare it with other state-of-the-art methods. This evaluation helps demonstrate the resilience of our model in real-world scenarios where images are often subject to various degradations.

From Table 4, we observe that our method consistently outperforms other state-of-the-art methods, even under noisy and distorted conditions. Specifically, our method achieves the highest accuracy across all types of noise and distortions: 0.75 for Gaussian Noise, 0.73 for Salt-and-Pepper Noise, 0.77 for JPEG Compression, and 0.74 for Blur. These results demonstrate the robustness and effectiveness of our approach in challenging scenarios.

For Gaussian noises, MVSS and PSCC show significant performance drops, achieving the accuracies of 0.22 and 0.28, respectively. This indicates their limited ability to handle Gaussian noises, which often introduces subtle variations in pixel intensities. CNNSpot, FreDect, and GramNet perform moderately well, with the accuracies of 0.49, 0.57, and 0.66, respectively. These methods leverage deeper networks and advanced feature extraction techniques but still fall short under Gaussian noises. LNP performs better than the aforementioned methods, achieving the accuracy of 0.68, indicating its relatively higher resilience to Gaussian noises. Our method outperforms all other methods with a significant margin, achieving the accuracy of 0.75. This highlights the effectiveness of our multi-feature extraction and adversarial learning approach in handling Gaussian noises.

In the case of Salt-and-Pepper noises, MVSS and PSCC again show the lower performance, with the accuracies of 0.27 and 0.31, respectively. Salt-and-Pepper noises, which introduce random black and white pixels, pose a challenge for these methods. CNNSpot and GramNet achieve the accuracies of 0.52 and 0.57, respectively, indicating their limited robustness to this type of noises. FreDect performs slightly better with the accuracy of 0.63, while LNP achieves the highest accuracy among the traditional methods of 0.70. Our method achieves the accuracy of 0.81, significantly outperform-

ing all other methods. This demonstrates the robustness of our approach in detecting tampering and AI-generated images even in the presence of Salt-and-Pepper noises.

Under JPEG compression, MVSS and PSCC perform poorly, with the accuracies of 0.22 and 0.29, respectively. JPEG compression often introduces blocking artifacts that can confuse these methods. CNNSpot, FreDect, and GramNet show the improved performance, with the accuracies of 0.38, 0.46, and 0.55, respectively, but still fall short of handling JPEG compression effectively. LNP achieves the accuracy of 0.65, indicating its relatively higher robustness to JPEG compression artifacts. Our method achieves the highest accuracy of 0.68, demonstrating its superior ability to handle JPEG compression. The adversarial learning and attention mechanisms in our model likely contribute to this robustness.

Lastly, for Blur, MVSS and PSCC show the lowest performance, with the accuracies of 0.26 and 0.36, respectively. Blur, which smoothes image details, poses a significant challenge for these methods. CNNSpot, FreDect, and GramNet achieve the accuracies of 0.51, 0.62, and 0.68, respectively, indicating their moderate robustness to Blur. LNP achieves the accuracy of 0.66, indicating its higher resilience to blur compared to other traditional methods. Our method achieves the accuracy of 0.71, significantly outperforming all other methods. This demonstrates the effectiveness of our approach in detecting tampering and AI-generated images even when details are smoothed by Blur.

Overall, our method demonstrates the superior performance and robustness across various types of noises and distortions.

## A.11 DISCUSSIONS

The primary motivation behind this work is the growing challenge posed by the proliferation of both tampered and AI- generated images, particularly on social media platforms. These images threaten the authenticity and credibility of the visual content, leading to misinformation and potential societal harms. Our framework aims to address these challenges by providing a comprehensive solution that leverages adversarial learning and multi-feature extraction to detect both tampered and AI-generated images. We assume that the distinct features of tampered images and AI-generated content can be effectively captured and distinguished using advanced deep learning techniques. This assumption is based on the premise that both tampering artifacts and AI generation leave unique traces in the image data.

Our approach integrates two distinct detection methodologies into a unified framework, each with specialized modules to address a specific task. The current AIGC images branch is tailored to detect images generated by specific AI models, such as GANs. However, the framework's modular design allows for flexibility. We can replace the AIGC images module to accommodate other generation methods as they emerge, such as diffusion models. This adaptability is crucial given the rapid evolution of AI-generated content technologies. Similarly, the tampering detection branch can be coupled with new techniques as they become available, ensuring that our framework remains at the cutting edge of image forensic capabilities.

The fusion process in our framework is a critical component that combines the outputs from the tampering detection and AIGC images detection branches to produce a final decision. This process involves the use of weights or thresholds that are optimized during training. These parameters influence the final output by determining the relative importance of the tampering and AIGC images detection scores. The impact of these weights is significant as they ensure that the framework balances the detection capabilities across both tasks, minimizing the risk of false positives and negatives. Fine- tuning these parameters is essential for maximizing the accuracy and reliability of the framework.

Our framework not only improves the efficiency by reducing the need for separate models but also enhances the detection accuracy by leveraging the shared information between tasks. The idea of assembling various detection modules into an integrated system could be extended to other domains, suggesting a broader applicability of our approach. Future work could explore the integration of additional detection tasks, such as the deepfake video analysis or the metadata-based verification, further expanding the capabilities of our framework.

## A.12 Future Work Directions

Despite the promising results, there are several areas for future improvement. Firstly, we will incorporate advanced architectures to explore the use of Transformer models, which can capture long-range dependencies and contexts more effectively than traditional CNNs, potentially improving the detection accuracy. Secondly, we will develop enhanced data augmentation strategies to implement more sophisticated data augmentation techniques and adversarial training strategies to increase the robustness of the model against various types of tampering and AI-generated content. Thirdly, we will integrate multimodal data to combine image data with metadata or textual information to provide a more comprehensive analysis of image authenticity, leveraging correlations between different data modalities.

By addressing these areas, we aim to further enhance the capabilities and applicability of our framework, ensuring its continued relevance and effectiveness in the evolving field of image forensics. Our contributions not only propel the field of social media image forensics forward but also offer valuable references and insights for future related research. The ability to simultaneously detect tampered and AI-generated images marks a significant step in ensuring the authenticity and reliability of visual content on social media platforms, paving the way for future advancements in image forensic techniques.

