# OpenReview forum: "Advanced Image Forensics: Detecting Tampered and AI-Generated Images with Adversarial Learning"
_ICLR.cc/2026/Conference — Submitted to ICLR 2026_

### Official Review · Reviewer_3Ufx · 2025-10-20

**Soundness:** 2
**Presentation:** 2
**Contribution:** 2
**Rating:** 2
**Confidence:** 4

**Summary:**

The paper claims to employ adversarial learning with a generator \( G \) and three discriminators \( D \), where \( G \) aims to produce an image \( \hat{I}_t \) that appears "real" to deceive the discriminators.

**Strengths:**

please refer to the "Questions" sections directly.

**Weaknesses:**

please refer to the "Questions" sections directly.

**Questions:**

- In image forensics—e.g., as in [1]—generators are typically used to simulate forgery processes, thereby enriching the training data for discriminators with diverse synthetic manipulations. In contrast, the generator here appears to "repair" tampered or AI-generated inputs by transforming them into realistic-looking images. This approach contradicts the core objective of forgery detection, which is to learn discriminative features of manipulation rather than to synthesize or restore realistic content. Consequently, the framework’s motivation is unclear, and its effectiveness for detection tasks is questionable.

[1] Zhuo et al., Self-Adversarial Training Incorporating Forgery Attention for Image Forgery Localization, IEEE TIFS, 2022.

- The paper proposes a unified framework for detecting both tampered and AI-generated images. While the goal is commendable, several key aspects require clarification:

a) Ambiguous Detection Target: The introduction highlights "Tampered AIGC" as a distinct category requiring detection. However, it's unclear whether the model's target for such images is to identify them as AI-generated, as tampered, or as a combined state. The output lacks explicit indicators for which specific task (tampering vs. AI-generation) the detection applies to, making the "simultaneous detection" objective ambiguous.

b) Unclear Labeling for Tampered AIGC Dataset: Table 2 presents pixel-level metrics (ACC, F1) for the "Tampered AIGC" dataset. It is crucial to clarify whether the ground truth labels for this dataset are at the image level (indicating if the image is tampered/AI-generated) or pixel level (marking specific tampered regions). This directly impacts the interpretation of the reported results.

c) Inconsistent Metric Interpretation in Table 1: Table 1 reports ACC and F1 scores across diverse datasets (tampered, GAN-generated, combined). Given that Section 4 specifies pixel-level evaluation for tampered regions and image-level for AI-generated content, it is ambiguous whether the F1/ACC values in Table 1 refer to pixel-level or image-level performance for each dataset type. This inconsistency hinders clear comparison.

d) Lack of Visual Examples: The paper mentions applying "adversarial tampering" to create the Tampered AIGC dataset but provides no visual examples. Including sample images would significantly aid readers in understanding the nature and subtlety of these specific manipulations.

- The paper aims to address the joint detection of both tampered and AI-generated images. However, the evaluation is limited to only two tampered datasets (CASIA V2 and IDM) and three GAN-based synthetic datasets (ProGAN, StyleGAN, CycleGAN). Given that AI-generated images now encompass diverse generation paradigms—especially diffusion models—the exclusive focus on GAN-generated content undermines the generalizability of the proposed method.
To strengthen the evaluation, the authors may align their test sets with those used in recent state-of-the-art methods such as TruFor [2], which includes comprehensive tampered benchmarks (e.g., Columbia, Coverage, NIST, DSO) and modern AI-generated image datasets like GenImage [3], covering a broader spectrum of generative models.

[2] Guillaro et al., TruFor: Leveraging All-Round Clues for Trustworthy Image Forgery Detection and Localization, CVPR 2023.

[3] Zhu et al., GenImage: A Million-Scale Benchmark for Detecting AI-Generated Images, NeurIPS 2023.

- Other issues:

(a) Table 4 does not specify the evaluation metric used, making the reported values ambiguous.

(b) The decision threshold for computing F1-scores in Tables 1–3 is not stated.

(c) While Appendix A.7.1 and A.7.3 define the pixel-level F1-score, the image-level F1 and accuracy (ACC) metrics referenced in Section 4 are not formally described. Providing their definitions would improve reproducibility and clarity.

(d) Table 1 is not referenced in the main text.

(e) The methods “TruFor” and “DFVT” listed in Table 1 are not discussed or cited anywhere in the manuscript, and “DFVT” corresponding references are missing.

---

### Official Review · Reviewer_kvhH · 2025-10-27

**Soundness:** 1
**Presentation:** 1
**Contribution:** 1
**Rating:** 2
**Confidence:** 5

**Summary:**

This paper proposes a joint image forensics task by integrating tamper detection and AIGC detection, and presents a simple model with two separate branches for tamper detection and AIGC detection respectively. The model incorporates adversarial training to improve detection accuracy. Finally, a small mixed dataset containing both tampered and AIGC images is constructed, and the model demonstrates certain performance advantages on this dataset.

**Strengths:**

- Proposes a joint image forensics task by integrating tamper detection and AIGC detection.
- Presents a joint detection model for tamper and AIGC based on adversarial training.
- Some experimental results show improvements compared with existing works.

**Weaknesses:**

- Unreasonable motivation explanation: The paper states that "Existing solutions often focus on either tampered or AIGC images, yet both can coexist in various contexts", but fails to explain why tamper detection and AIGC detection need to be addressed jointly. What is the correlation between them? If tampering mainly refers to manual Photoshop manipulation, its artifacts and the generative artifacts of AIGC belong to two different categories. Why is it beneficial to learn them simultaneously? It is suggested that the authors conduct a more in-depth analysis of the problem background. Additionally, simply combining the two categories is not sufficient to support it as a contribution.
- Insufficient technical innovation in the method: First, the use of adversarial learning to improve robustness/generalization is not novel. Moreover, the network modules, loss functions, and other components covered in the method section are all based on existing works. What is the authors' core original contribution?
- The constructed dataset (Tampered AIGC images Dataset) lacks novelty: It is simply created by applying copy-move, splicing, inpainting, and other tampering operations to 10,000 AIGC images. On the one hand, the workload involved is relatively simple. On the other hand, compared with existing datasets, it does not demonstrate any unique attributes.
- Unclear experimental setup: It is impossible to determine which training set the authors used. Furthermore, in Table 1, comparative methods such as TruFor and CNNSpot were trained on different training sets, making it unfair to compare them directly.

**Questions:**

See weaknesses.

---

### Official Review · Reviewer_xb9y · 2025-10-31

**Soundness:** 1
**Presentation:** 1
**Contribution:** 1
**Rating:** 2
**Confidence:** 5

**Summary:**

This paper introduces a framework for unified image forensics, designed to simultaneously detect both tampered images and AI-generated content (AIGC). The core of the proposed method is an adversarial learning architecture featuring a generator and a tri-branch discriminator. The three discriminator branches are specialized to identify distinct forensic traces: one for edge-based tampering artifacts, one for noise-based inconsistencies, and a third for features characteristic of AI-generated images.
Through adversarial training, the generator learns to produce increasingly realistic forgeries that challenge the discriminator, which in turn enhances the discriminator's ability to detect subtle manipulations. To facilitate evaluation, the authors construct a new benchmark dataset named "Tampered AIGC images Dataset," which contains AI-generated images that have been subsequently manipulated.
The experimental results demonstrate that the proposed framework significantly outperforms existing aigi detection and imdl methods across various datasets, including standard tampering benchmarks (CASIA V2, IDM), GAN-generated datasets (ProGAN, StyleGAN, CycleGAN), and their custom combined dataset.

**Strengths:**

The primary merit of this paper lies in its focus on a highly relevant and pressing issue. The authors correctly identify the need for a unified framework that can simultaneously detect traditional image tampering and AI-generated content, a scenario that is increasingly common in real-world applications like social media analysis.

**Weaknesses:**

Severely Flawed and Incomprehensible Methodological Description: The paper's core methodology, particularly the formulation of its loss functions and the adversarial training procedure, is riddled with inconsistencies, ambiguities, and conceptual errors. This makes the proposed method impossible to understand or reproduce, and it casts serious doubt on the validity of the implementation used to generate the results. For example, the authors define the adversarial loss L_adv in two conflicting ways. Equation (2) presents the standard GAN minimax objective, while Equation (12) (and Algorithm 1) redefines L_adv as a weighted sum of the discriminators' Binary Cross-Entropy (BCE) losses.

The authors seem to have a severe disconnect between their architectural diagram, their mathematical formulation, and their actual experimental implementation. They present the AIGC branch as a segmentation/localization module but appear to have trained and evaluated it as a simple image classifier.

Outdated and Incomplete Experimental Scope: The paper's evaluation of AIGC detection is exclusively focused on GAN-based models (ProGAN, StyleGAN, CycleGAN). For a submission targeting ICLR 2026, the complete absence of experiments on diffusion models (e.g., Stable Diffusion, Midjourney)—the dominant class of generative models for the past several years—is a critical oversight.

Unreasonable experimental setup: As a method that integrates AIGI and IMDL detection, the paper retrains other AIGI and IMDL baselines on mixed datasets for comparison, causing those methods to lose their basic domain-specific capabilities. For example, on GAN-generated datasets, methods like CNNSpot should achieve 90%+ accuracy. This undermines fairness. In addition, the paper lacks a basic comparison that fuses two domain-specific detectors via a combined decision, which makes its fusion strategy unconvincing and raises doubts about whether the problem formulation is meaningful in the first place.

**Questions:**

Please refer to "Weaknesses" for details.

---

### Official Review · Reviewer_3RVG · 2025-11-01

**Soundness:** 3
**Presentation:** 2
**Contribution:** 2
**Rating:** 4
**Confidence:** 3

**Summary:**

The paper proposes a dual-branch network that jointly performs AIGC detection and tampering localization via cross-attention and feature fusion. A new Tampered AIGC Dataset is introduced, built by applying copy-move, splicing, and inpainting to ProGAN/StyleGAN/CycleGAN outputs. The method outperforms existing forensic baselines, achieving 79% accuracy and 0.76 F1-score on a mixed-domain test set.

**Strengths:**

- The technical approach is well-motivated. The dual-branch architecture with cross-attention enables mutual learning between global authenticity cues (AIGC vs. real) and local inconsistency signals (tampering boundaries).
- Paper is clearly written and easy to follow.

**Weaknesses:**

- Dataset uses GANs (ProGAN/StyleGAN) but omits modern diffusion-based generators (e.g., Stable Diffusion), limiting practical relevance.
-  Edits are traditional (copy-move, etc.), not reflecting current AI-assisted editing tools (e.g., Generative Fill and Inpainting).
- Misses comparison to recent AIGC-aware forensics (e.g., UniversalForensics, GenImage).
- Lacks localization metrics (e.g., IoU)

**Excessive length: The paper exceeds the 9-page limit.**

**Questions:**

Please refer to Weaknesses.

---

### Meta-Review · Area_Chair_A8Qq · 2026-01-08

**Summary:**

The reviewers has consensus that the current submission falls short in soundness, clarity, and substantiated contribution. Across reviews, concerns are raised regarding the conceptual motivation for jointly modeling these tasks, internal inconsistencies between the architectural design, mathematical formulation, and claimed objectives, and lack of clarity. Multiple reviews identify fundamental ambiguities in the loss definitions and training procedure that undermine reproducibility and confidence in the reported results. From an experimental perspective, the exclusive focus on GAN-based generators and traditional edits is broadly viewed as outdated and insufficiently representative. Limited evaluation and missing comparisons to recent state-of-the-art forensic methods further weakening the empirical case.

No author rebuttal was provided, leaving substantive reviewer questions unanswered and removing any opportunity to resolve or mitigate these weaknesses. The justified recommendation is rejection.

**Reviewer Concerns:**

There is no rebuttal provided.

**Reviewer Scores:**

All reviewers would likely maintain or lower their scores if given full discussion. Reviewers 2 and 3 would stay firm on rejection due to fundamental flaws, Reviewer 4 would likely solidify a reject, and Reviewer 1 might move from borderline to a clearer reject without author clarification.

---

### Decision · Program_Chairs · 2026-01-26

Reject